# Fruit and Vegetable Incentive Programs for Supplemental Nutrition Assistance Program (SNAP) Participants: A Scoping Review of Program Structure

**DOI:** 10.3390/nu12061676

**Published:** 2020-06-04

**Authors:** Katherine Engel, Elizabeth H. Ruder

**Affiliations:** Rochester Institute of Technology; Rochester, NY 14620, USA; katherine.engel4@gmail.com

**Keywords:** incentive programs, Supplemental Nutrition Assistance Program (SNAP), fruits and vegetables, low-income, farmers’ markets, dietary quality, produce intake, produce purchasing

## Abstract

The low intake of fruits/vegetables (FV) by Supplemental Nutrition Assistance Program (SNAP) participants is a persistent public health challenge. Fruit and vegetable incentive programs use inducements to encourage FV purchases. The purpose of this scoping review is to identify structural factors in FV incentive programs that may impact program effectiveness, including (i.) differences in recruitment/eligibility, (ii.) incentive delivery and timing, (iii.) incentive value, (iv.) eligible foods, and (v.) retail venue. Additionally, the FV incentive program impact on FV purchase and/or consumption is summarized. Using the Preferred Reporting Items for Systematic Reviews and Meta-Analyses (PRISMA) guidelines for scoping reviews, a search of four bibliographic databases resulted in the identification of 45 publications for consideration; 19 of which met the pre-determined inclusion criteria for full-length publications employing a quasi-experimental design and focused on verified, current SNAP participants. The data capturing study objective, study design, sample size, incentive program structure characteristics (participant eligibility and recruitment, delivery and timing of incentive, foods eligible for incentive redemption, type of retail venue), and study outcomes related to FV purchases/consumption were entered in a standardized chart. Eleven of the 19 studies had enrollment processes to receive the incentive, and most studies (17/19) provided the incentive in the form of a token, coupon, or voucher. The value of the incentives varied, but was usually offered as a match. Incentives were typically redeemable only for FV, although three studies required an FV purchase to trigger the delivery of an incentive for any SNAP-eligible food. Finally, most studies (16/19) were conducted at farmers’ markets. Eighteen of the 19 studies reported a positive impact on participant purchase and/or consumption of FV. Overall, this scoping review provides insights intended to inform the design, implementation, and evaluation of future FV incentive programs targeting SNAP participants; and demonstrates the potential effectiveness of FV incentive programs for increasing FV purchase and consumption among vulnerable populations.

## 1. Introduction

Eating sufficient amounts of fruits/vegetables (FV) is vital for a healthy dietary pattern associated with a lower risk of cardiovascular disease and certain cancers [1]. However, Americans do not consume enough FV; only 12.2% and 9.3% of US adults meet the *2015–2020 Dietary Guidelines for Americans*’ recommendations for daily fruit and vegetable consumption, respectively [2]. Among Americans, lower income groups consume less FV than higher income groups, and this is a key socioeconomic disparity in overall dietary quality [2,3,4]. Thus, it is important that low-income participants in federal food assistance programs in the United States, such as the Supplemental Nutrition Assistance Program (SNAP), have access to these foods.

SNAP is the largest federal food assistance program in the United States. It functions by providing participants with food purchasing resources in the form of an electronic benefit transfer (EBT, an electronic system that allows a recipient to authorize transfer of their government benefits from a federal account to a retailer) on a monthly cycle. Although SNAP eligibility requirements vary from state to state, households that are SNAP eligible have gross incomes of less than 130% of the federal poverty line [5]. Unlike other U.S. food assistance programs, like the Supplemental Nutrition Assistance Program for Women, Infants, and Children (WIC), SNAP benefits can be used for most food products with few exceptions (such as hot foods and foods that are intended to be eaten in stores) [6,7]. In contrast to SNAP, WIC benefits are limited to foods such as milk, cheese, yogurt, FV, canned fish, tofu, breakfast and infant cereal, whole wheat breads and grains, eggs, peanut butter, infant formula, and jarred baby foods [8]. Thus, although SNAP plays an integral role in ensuring that millions of people have the resources they need to access sufficient amounts of food, it lacks specific restrictions that dictate the nutritional quality of foods that participants can purchase. Importantly, it has been shown that individuals who receive SNAP benefits have poor diets relative to the overall population and other income-eligible non-participants [3]. In some cases, SNAP participation has been associated with negative health outcomes and inversely correlated to self-assessed health status [9]. Given the evidence that WIC participation is associated with health benefits [10], one proposed alteration to the SNAP program is creating restrictions around which foods can be purchased with benefits. However, key constituencies, ranging from members of U.S. Congress to hunger relief organizations, have rejected these proposals for reasons including concerns about limiting participants’ ability to exercise autonomy in food choice and administrative burdens [11]. Moreover, restrictions on the types of food eligible for SNAP could contribute to worsening food security in areas where a variety of healthful foods is not sold by food retailers. Another alteration to the SNAP program that has been suggested is FV incentives, which provide participants with considerable autonomy in deciding what foods to purchase [12].

FV incentives include a variety of inducements to offer low-income participants funds to purchase these foods. They are potentially appropriate for improving dietary quality, because they are a tool for facilitating behavior change. The theory that incentives serve as a strategy for inducing changes in behavior centers on the standard direct price effect [13]. The standard direct price effect makes the incetivzed behavior more attractive by providing a financial reward. As a result, incentives have the capacity to instill new, positive habits, as well as end pre-existing, negative habits. Thus, when applied on a large enough scale, incentives may have the ability to shift cultural norms [13]. Incentives may be particularly useful for promoting healthy behaviors, such as consuming more FV, because the benefits of healthy behaviors are often uncertain and delayed, while the cost of these behaviors is immediate. Consumers tend to value current costs and benefits more than future costs and benefits, which in turn can lead to choosing not to engage in healthy behaviors, since the present value of these behaviors is low. Incentives create an immediate benefit because they lower the cost of healthy foods for consumers. By creating short-term rewards for healthy behaviors, incentives serve to make these behaviors more appealing by increasing their present value [14]. In general, the cost of food plays a critical role in how people make food choices [15,16]. Glanz and colleagues [17] found that behind taste, price is the second most important influence on food choice. For SNAP participants specifically, it has been demonstrated that the cost of healthy foods is a barrier for improving dietary quality [18,19]. Incentives expand the financial resources participants have available to purchase healthy foods, and thus address the barrier that the cost of these food poses to dietary quality [18,19,20].

FV incentive programs have been designed and implemented for a number of different populations, including WIC and SNAP participants, and venues, such as farmers’ markets and grocery stores. In addition, the types and value of incentives that have been developed vary widely from point-of-sale (POS) discounts to coupons, vouchers, and tokens. A preliminary search for existing scoping reviews on this topic was conducted by searching the Cochrane Database of Systematic Reviews, Google Scholar, ProQuest, PubMed, and Sage Journals Online. No scoping reviews on this topic were identified. Given the emerging evidence related to FV incentive programs among SNAP participants and the diversity of structural factors within these programs, a scoping review was selected as the appropriate method. The objective of this scoping review is to characterize the factors in program structure which may impact the effectiveness of incentive programs. The scoping review research question is, “What are the differences in structural factors, including recruitment and eligibility criteria, delivery and timing of incentives, financial value of incentives, foods eligible for incentive redemption, and type of retail venue reported among FV incentive programs?” Finally, this review summarizes the outcomes of existing FV incentive programs with respect to the purchase and/or consumption of FV among SNAP participants, with specific attention to the quality of the assessment methods for FV purchase and/or consumption. This work provides insight intended to inform the design, implementation, and evaluation of future FV incentive programs targeting SNAP participants.

## 2. Materials and Methods

A scoping review was undertaken to systematically synthesize factors in program structure which may impact the effectiveness of FV incentive programs. This review was conducted as per the Arksey and O’Malley framework for scoping reviews [21] and integrated with the guidance from the Joanna Briggs Institute (JBI) [22] and the Preferred Reporting Items for Systematic Reviews and Meta-Analyses (PRISMA) guidelines for scoping reviews [23]. A protocol document is publically available online at: https://figshare.com/articles/Protocol_Document_pdf/12380669, and a completed PRISMA ScR checklist is included as Appendix A.

### 2.1. Search Strategy

Focused searches were conducted by one author (K.E.) using Google Scholar, ProQuest, PubMed, and Sage Journals Online. The search terms that were used include “SNAP incentives,” “WIC incentive,” “food benefits incentive,” and “food assistance incentive.” Results were limited to English language publications and indexed up to 7 November 2019. In addition to the use of these search terms, papers were identified by examining the articles cited by the papers found in the preliminary search.

### 2.2. Study Selection

Full-text articles identified in the search were imported into Mendeley reference management software and duplicates were manually removed. In total, 45 unique publications were identified and both authors independently reviewed the full-text documents for pre-determined inclusion/exclusion criteria. The inclusion criteria included: full-length publication in a peer-reviewed journal or government report, quasi-experimental design, and targeted focus on verified, current SNAP participants studies that solely examined the use of FV vouchers as part of the WIC foods package were excluded for the following reasons: (1.) FV vouchers became a standard part of the WIC Food Package following a final rule published in May 2014 [24] and (2.) WIC FV vouchers can only be used for FV and therefore are not used to incentivize the purchase of FV over other foods within the WIC Food Package. The authors conferenced regularly to ensure agreement and talked through any inconsistencies. Due to the relative lack of research on this topic, papers were not excluded based on their publication date.

### 2.3. Data Charting

Data were extracted from eligible papers into a standardized Google Doc chart developed by both authors. The two authors independently charted the data, discussed the results and continuously updated the data collection chart. The data ultimately collected included: study authors, year of publication, study objective, population and sample size, methodology, incentive program structure characteristics (participant eligibility and recruitment, delivery and timing of incentive, foods eligible for incentive redemption, type of retail venue), and study outcomes related to FV purchases and FV consumption. In addition, study methods for the assessment of FV purchase and/or consumption were charted. The charted data were summarized as counts where applicable.

## 3. Results and Discussion

Of the 45 publications initially reviewed, *n* = 6 were excluded for not falling within the scope of the review, *n* = 8 were excluded for not employing a quasi-experimental design, *n* = 6 were excluded because participation was not focused on current, verified SNAP participants, and *n* = 2 were excluded for being solely related to WIC FV vouchers prior to the implementation of the 2014 WIC Food Package. In addition, two poster presentations were excluded and two publications were excluded because they presented preliminary data that was included in a subsequent publication. In total, 19 publications were included in the final review (Table 1).

### 3.1. Incentive Program Structure

A variety of types of incentive programs have explored approaches for increasing the purchase and consumption of FV by SNAP participants. The following section details ways in which eligible individuals become participants in incentive programs, the delivery and timing of incentives, and differences in the financial value of incentives to participants.

#### 3.1.1. Recruitment and Eligibility of Incentive Program Participants

In the studies under review, individuals became incentive program participants in a multitude of ways. Eleven programs had an enrollment process through which individuals had to complete some type of informal or formal sign-up process for the program to receive the incentive [7,26,27,28,31,32,33,36,38,40,41], while eight studies had no enrollment process and provided the incentive when participants visited and/or made a purchase at a retailer and provided evidence of their SNAP participation [25,28,30,34,35,37,39,42]. Bowling et al. provided all SNAP participants shopping at participating markets with an incentive and provided an additional incentive to a subset of this population that had specifically enrolled in the program [27]. It is important to note that the inclusion of an enrollment process may create additional administrative challenges, as well as barriers for participation. However, as enrollment processes often included a pre-test survey and/or a method of tracking participants’ transactions throughout the implementation period, these programs may provide opportunities for more rigorous evaluation and therefore greater insight regarding the impact of incentive programs of FV purchases and consumption. One study assessed programs in which participants were given the incentive after visiting a health clinic [33]. Similar to the challenges with enrollment processes, this requirement may create a participation barrier, but may have a greater impact on dietary quality and health, as it is part of a broader focus on the health status of federal food assistance program participants.

#### 3.1.2. Delivery and Timing of Incentive Benefits

Table 1 summarizes the types of incentive benefits that have been granted to participants. Two programs were structured such that the incentive was provided at the point-of-sale (POS) [7,31]. For the purposes of this review, POS incentives are defined as those that immediately discount participants’ FV purchases at checkout. In contrast to this model, 17 programs provided participants with coupons, vouchers, or tokens [25,26,27,28,29,30,32,33,34,35,36,37,38,39,40,41,42]. The delivery of coupons, vouchers, or tokens (hereafter referred to as incentives) varied by program. Some programs provided the incentives when the participant enrolled in the program or following their enrollment, such as when they visited a farmers’ market [28,32,35,41]. Other programs provided the incentives following or in conjunction with the purchase of FV [7,25,26,28,30,31,33,36,37,38,40]. Moreover, some programs required incentives to be redeemed immediately upon receipt [7,31], but others allowed the incentive to be used for a future transaction. Importantly, allowing participants to save the incentive and choose when they use it may be beneficial, due to the monthly “SNAP-cycle” spending pattern, where the majority of recipients spend most of their monthly benefits within two weeks after receiving them [43,44]. In all cases, the intent of these benefits is to induce participants to increase their FV purchases by providing them with financial rewards for these purchases and/or resources that enable them to purchase these foods at a lower price.

Most incentive programs included in this review required participants to make an FV purchase in order to “trigger” the delivery of the incentive benefit [7,25,26,28,30,31,33,36,37,38,40]. However, the types of FV that qualify as trigger foods differ. For example, the Healthy Incentives Pilot, a federally funded FV incentive program administered in Hampden County, MA, distributed incentive benefits after participants purchased targeted FV, which were defined as any fresh, canned, frozen, and dried fruit or vegetable FV without any added sugars, fats, oils, or sodium. In addition, the pilot excluded fruit juice, mature legumes, and white potatoes. These specifications were selected to mirror the restrictions of WIC-eligible produce items [7]. In contrast, for programs held at farmers’ markets, participants often had to purchase fresh FV in order to receive the incentive.

A few programs had multiple points and locations at which incentive benefits were distributed to participants. In the program evaluated by Young et al. [42], a $2 bonus incentive coupon was provided for every $5 in SNAP benefits used at a farmers’ market. Additionally, coupons were distributed at community organizations that serve SNAP-eligible populations, absent of any initial purchase by the participant. The program examined by Olsho et al. [34], was structured similarly, in that some participants received the incentive through a match after they made a purchase, while others received the incentive from community-based organizations absent of any purchase, usually after they attended a nutrition workshop or other health and fitness program. Similarly, two incentive distribution methods were employed in the program examined by Savoie-Roskos et al. [39]; participants received either “regular incentives”, which were distributed at regular intervals without any purchase requirement, or matched incentives. Bowling et al. [27] employed both POS incentives and tokens; all SNAP recipients shopping at participating markets received a matched incentive when they used their EBT card at these markets, which could not be saved for future use, but at every third market, participants also received $20 in “Bonus Buck” tokens.

#### 3.1.3. Financial Value of Incentive to Participants

The value of incentive benefits to participants differed widely. As stated previously, most programs required a purchase to receive the incentive [7,25,26,27,30,31,32,34,36,37,38,40,42]. In these programs, the value of the benefit was either pre-determined or determined by the value of the participants’ purchases. For example, in the incentive program studied by Freedman et al. [31], participants received benefits valued at $5 regardless of the cost of their initial purchases. However, many incentive programs functioned such that the value of the benefit was determined by the magnitude of the participants’ spending [7,25,26,27,29,30,32,34,36,37,38,39,40,42]. In these cases, the value of the benefit was either equal to the participants’ spending or a percentage of their spending. In many instances, 100% of the participants’ spending was matched, meaning that the value of the benefits was equal to the amount of money spent by participants [25,26,30,32,36,37,38,39,40]. In some cases, the value of the benefit was adjusted based on the size of participating families, as families with children were given additional value [38].

Among incentive programs that provided a match to participant spending, there was frequently a ceiling on the value of the match. In all, 11 of the studies [7,25,26,27,28,30,32,36,37,38,41] reviewed had some type of ceiling. For instance, in a Utah-based farmers’ market incentive program, participants received $1 in incentives for every $1 they spent in SNAP benefits, with individuals and couples receiving $10 worth of incentives each week and families receiving an additional $5 per child, up to $30 each week [39]. Another type of ceiling was demonstrated, where participants could receive an extra $20 in bonus tokens every third farmers’ market visit but were limited to receiving $120 of these bonus tokens during the program’s implementation [27]. Other programs provided benefits that were valued as a percentage of the participants’ spending. For instance, the Health Bucks and Philly Food Bucks programs provided $2 vouchers for every $5 participants spent, and thus acted as a 40% match of the participants’ spending [32,42,45]. Notably, incentives that are granted in proportion to the participants’ spending are designed to encourage participants to purchase more fruits and vegetables, because with these programs, the more participants spend on these foods, the more they are rewarded.

Some programs implemented multiple forms of incentives. For example, Savoie-Roskos et al. [39] provided one group of participants with incentive benefits that did not require them to make a purchase and another group with benefits, in the form of spending matches, that augmented the incentive. A comparison of the outcomes between the groups was not reported.

Overall, programs that match participants’ spending may provide incentive benefits that have greater financial value than those that provide a benefit of a fixed value. In addition, programs with ceilings may create less value for participants than those without ceilings. Thus, certain programs may be more effective in inducing participants to purchase and consume more FV, because they expand participants’ purchasing power to a greater degree. Differences in value may also be important, given that program retention is a challenge across the literature, and programs that provide less value may be less effective in encouraging ongoing participation. Notably, Wetherill et al. [41] posited that low incentive redemption rates may be tied to perceived differences in the value of different kinds of incentives, as incentives that function as discounts and expand buying power may be less valuable to participants than incentives that provide free products.

#### 3.1.4. Eligible Foods

The foods that were eligible for purchase using incentive benefits also differed. While some of the programs provided benefits that could be utilized to purchase only FV, others were triggered by an FV purchase, but provided benefits redeemable for a diverse range of foods, such as any SNAP-eligible food [7,36,39]. A drawback of awarding incentives that can be used for any SNAP-eligible food is the possibility that incentive benefits are used to purchase foods with low nutrient density. For example, in the Healthy Incentives Pilot, an additional $0.30 was added to participants’ EBT cards for every $1 of SNAP benefits spent on FV. There are few limitations on the types of foods and beverages that can be purchased with EBT, and some evidence suggests that reducing the price of healthful foods may result in the increased purchase of energy, which could contribute to obesity [46].

In contrast, other FV incentive benefits could be used only to purchase locally grown FV [27,29,31,32,34,37,40,47]. In some cases, the foods included in the incentive program varied based on participant eligibility. For example, in the Fresh Funds program [36], participants could use the tokens they purchased with their SNAP benefits, and the tokens that they received as a match, to purchase fresh produce or packaged foods, such as jams/spreads, breads, eggs, pasta, cheese, and fish; however, the tokens they purchased using WIC benefits could only be spent at vendors selling fresh produce.

The characteristics and needs of the recipients must be considered when designing incentive programs and the types of FV eligible for incentive redemption. For example, FV may not be an appealing incentive to people with limited facilities and equipment for food preparation. However, for participants with access to food preparation facilities, frozen, canned, and dried FV have a longer shelf-life than fresh FV and may be useful for prolonging food security throughout the month and between monthly SNAP benefit distributions. Likewise, the capacity of the retail environment must also be considered. SNAP vendor eligibility implemented in January 2018 requires vendors to stock FV, but does not require those FV to be fresh if other perishable foods are stocked (i.e., meat or dairy), and only one type of perishable food needs to be offered (i.e., selling just one type of fruit would fulfill the fresh FV requirement) [48]. Low income communities tend to have more convenience stores and small markets [49,50] where the availability of FV tends to be lower [51,52,53]. Therefore, the retail capacity, including the availability of freezers/refrigeration, must be considered when designing fresh FV incentive programs.

#### 3.1.5. Retail Venue

Table 1 illustrates that the majority (16/19) of the reviewed studies were implemented in part or in entirety at farmers’ markets. Farmers’ market incentive programs have the advantage of supporting local farmers and food vendors. Additionally, the literature indicates that shopping at farmers’ markets positively impacts FV purchases and that by drawing participants to shop at these venues, incentive programs implemented at farmers’ markets may positively impact FV purchase and consumption behaviors and attitudes beyond the time period in which the program is implemented [31,32,54].

Several studies indicate that farmers’ market incentive programs attract SNAP participants who otherwise might not shop at these venues [30,31,32,35]. One incentive program study found that 57% of participants in a farmers’ market incentive program had never been to a farmers’ market [31]. Similarly, another study noted that SNAP participants’ awareness of farmers’ markets rose in relation to their exposure to the Health Bucks incentive program [35]. In addition, these researchers found that 54% of Health Bucks participants who used their benefits at farmers’ markets strongly agreed that “I shop at farmers’ markets more often because of Health Bucks”, and a Utah-based incentive program reported that 98% of baseline participants reported that the incentive made it more likely that they would shop at the farmers’ market [30]. In the Farmers’ Market Fresh Fund Incentive Program, 82% of participants had never attended a farmer’s market prior to participating in the program, and 93% of participants reported that incentives were “important” or “very important” in their decision to shop at farmers’ markets [32]. In addition to drawing more SNAP participants to farmers’ markets, the Farmers’ Market Fresh Fund Incentive Program demonstrated the potential to impact participants’ long-term shopping behavior. In particular, the majority of participants reported that they would be “somewhat likely” or “completely likely” to shop at farmers’ markets even without the continuation of the incentive program [32]. Increased awareness that EBT is accepted at many farmers’ markets has also been noted among incentive program participants [39]. Accordingly, there is evidence that farmers’ market incentive programs increase participants’ exposure to markets as venues offering affordable, healthy food, and in turn have the potential to positively influence their long-term food purchasing behavior.

Another potential benefit of implementing incentive programs at farmers’ markets is the potential for the increased consumption of FV. Shopping at farmers’ markets is linked to increased FV consumption, and thus offering incentives at farmers’ markets has the capacity to improve dietary quality beyond merely increasing the financial resources participants have to purchase FV [55]. Specifically, Olsho et al. [34] found that both incentive program participants and farmers’ market shoppers who were not enrolled in the program reported higher FV consumption than other residents in their neighborhoods. However, incentive program participation per se was not related to an increase in daily FV servings.

Despite the potential benefits of implementing incentive programs at farmers’ markets, it is important to consider access issues in this context. Specifically, farmers’ markets are not as abundant as other types of food retailers, such as grocery stores, and may not exist in certain communities. Driving distance from residence to market has been inversely correlated with repeat use of farmer’s market incentives [56]. However, other research suggests that the distance from food retailers does not significantly affect the extent to which incentives impact SNAP participants’ FV spending [7,32]. Moreover, many markets are not open year-round and have limited hours of operation.

### 3.2. Outcome Assessment

All of the studies included in this review considered the impact of incentives on FV purchases and/or consumption. Four of the studies reviewed focused exclusively on FV purchases [25,30,36,39,41], five focused exclusively on FV consumption [28,29,34,37,40], and nine examined both FV purchase and consumption [7,24,26,27,31,32,33,35,38]. As explored later in this section, only one of the reviewed studies [41] did not report some positive impact on FV purchases and/or consumption in conjunction with incentive programs.

Studies employed a variety of approaches for measuring these outcomes, as shown in Table 2 and Table 3. The majority of studies assessed FV purchase and/or consumption using a pre-/post-test design, where participants’ FV purchase and consumption behaviors and attitudes were assessed prior to the implementation of the incentive program and then again at the program’s conclusion [25,26,27,28,30,31,32,33,34,35,36,38,39,42]. In addition, the Healthy Incentives Program also assessed the program impact at various points throughout the implementation phase [7]. A few studies also used control or quasi-control groups to assess program impact [7,34,37,41,47]. The merits of quasi-control groups are somewhat limited if the comparison groups do not share important characteristics with the incentive program participants. For example, Olsho et al. [34], compared the FV purchase and consumption of incentive program participants with that of other non-participant neighborhood residents, but these residents were not necessarily federal food assistance program participants. Although the groups may have shared relevant demographic characteristics, the comparison is problematic because federal food assistance program participants have unique circumstances that may make incentive programs particularly salient, such as the challenge of managing food-purchasing resources in conjunction with the monthly SNAP distribution cycle. Another study compared SNAP transaction data from participating grocery stores to that of nonparticipating stores to determine whether the percentage of dollars spent on fresh produce in total SNAP transactions is higher in stores that implement incentives than in stores that do not [37]. Control stores were selected using a coarsened exact matching and linear probability match to match on store characteristics and sociodemographics. While this approach is not as rigorous as randomizing stores to the incentive or control condition, the use of matched controls is preferred to non-matched controls. Wetherill et al. [41] employed a quasi-experimental design to compare two coupon interventions: basic information and plain coupon distribution compared to tailored, targeted marketing coupon intervention. However, low coupon redemption by either group made comparisons difficult.

Assessment of FV consumption varied in the quality of methods used for dietary assessment. Of the 15 studies which assessed the change in FV consumption (Table 3), five studies employed the validated Behavioral Risk Factor Surveillance System FV module [57], and two other studies used other validated assessment tools [7,29]. The validity of the dietary assessment methods for the remaining eight studies was not clear.

All studies under review noted some degree of positive impact, with the exception of Wetherill et al. [41]. In that study, participants were all recipients of Temporary Assistance for Needy Families (TANF), a cash-assistance program in the United States for very low-income families with children. Generally, TANF participation automatically qualifies a household for SNAP. Given the severe income restriction of TANF households, these participants may not be representative of the general SNAP population. Moreover, few participants in this study (*n* = 16, 6.3%) redeemed the incentive coupons; making outcome assessment difficult, although the authors did note that that education surrounding food preparation skills may be necessary, in conjunction with incentives to alter food purchasing behaviors at farmers’ markets. Among the remaining studies that reported some positive impact, limitations in the impact of the incentive programs were identified. Conclusions from the Michigan farmers’ market-based Double Up Food Bucks program included that the impact of incentive programs was unsustainable and minimal [40], and a Washington, DC-based farmers’ market evaluation noted that although participants reported higher FV consumption, their intake still fell below recommended levels [35]. Olsho et al. [34], reported an increase in purchases but concluded that there was no observable difference in consumption between incentive program participants and non-participants, and similarly, Bowling et al. [27] observed that while participants reported increased fruit and vegetable consumption, they did not change the amount of their WIC/SNAP budget spent on these foods.

Several demographic factors have been linked to incentive program retention and use frequency and thus may be important when considering outcome assessment. Specifically, Dimitri et al. [29] noted that participants who were more reliant on food banks, very income restrained, and lived in areas where access to food was limited were more likely to drop out of the incentive program they studied [29]. These findings suggest that the presence of these factors may impact the effectiveness of incentive programs, as participant retention is essential for incentives to influence FV purchases and consumption. Ratigan et al. found that participants who had unhealthier diets at the beginning of the program were more likely to use incentives a greater number of times in the short term, but incentive use waned after six months [36]. In addition, Ratigan et al. noted that elderly and disabled individuals were more likely to use incentive programs in the long term than those who were younger and noted that ethnicity, type of government food assistance program participation, income, season of incentive program enrollment, and baseline FV consumption were related to the frequency of incentive utilization and total duration of their retention in the incentive programs [36]. They also noted that ethnicity, type of government food assistance program participation, income, season of incentive program enrollment, and baseline FV consumption correlate with both the number of times participants utilize incentives in a given period of time, as well as the total duration of their retention in incentive programs. Together, these results suggest that additional work is needed to identify the characteristics of subgroups who are most responsive to incentive programs in order to target incentive programs.

## 4. Conclusions and Recommendations

This scoping review highlights the wide range of FV incentive program structures and demonstrates that, in general, these programs may be an effective approach for increasing FV purchase and consumption by SNAP participants, while preserving autonomy in food choice. However, it is unclear whether the potential positive effects of these programs are substantial and sustainable. Moreover, the assessment methods employed to evaluate these programs have often relied on self-reports and lack sufficient rigor to assess program impact. Specifically, dietary assessment, when performed, frequently failed to utilize validated methods, such as the 24-h recall. Additionally, there are limitations to the scoping review process itself. Namely, a professional librarian was not consulted to assist with developing the search strategy and the protocol was not published early enough in the process to allow input from the greater scientific community. These factors, in conjunction with the significant variation in program structure, makes it difficult to elucidate which programmatic elements may be most critical for designing and implementing effective programs.

Although the literature indicates that incentive programs may positively impact FV purchases and consumption by SNAP participants, several areas that require additional research in order to understand how to create effective programs are revealed. For instance, other interventions, such as nutrition education, cooking demonstrations, and food tastings, are often deployed in conjunction with incentives. These interventions not only equip participants with the knowledge they need to make healthy eating decisions and integrate healthy foods into their diets but may also contribute to participant use and retention. Participant use of available incentives and retention is a key determinant of program effectiveness, and additional research is needed to understand how to maximize participation and retention. Additional work is needed to elucidate how participant characteristics, such as food security status and demographics, may be associated with the use of incentives. Another area for future research is the impact of incentive program participation on objective measures of health. The studies reviewed in this scoping review demonstrated an improvement in program participants’ perceptions of their health [7,32,33,36] and one study demonstrated an improvement in food security status [30], but none of the studies under review measured BMI or other health measures. Consequently, the actual impact of incentives on health remains unclear, and identifying the point at which incentives create a tangible difference in health outcomes is key for creating programs that promote participants’ well-being. Lastly, more research is needed to understand the long-term effects of incentives. Some evidence suggests that the increases in FV purchases resulting from incentive program participation are not sustained following program termination [40]. Additionally, few studies have investigated the capacity of incentive programs to influence long-term food consumption and purchasing behavior. Thus, the long-term efficacy of incentives is uncertain [38,58]. Moreover, the research that has considered the long-term impacts of incentive programs has relied on self-reported predictions of future food purchasing behavior [32]. As no longitudinal studies of the impact of incentive programs have been performed, additional research is required to determine the long-term impact of these programs.

Overall, studies of FV incentive programs reveal a positive impact on both FV purchases and consumption. This scoping review provides insights intended to inform the design, implementation, and evaluation of future FV incentive programs targeting SNAP participants. Exploring these factors is critical for understanding how to effectively design and implement effective, sustainable incentive programs.

## Figures and Tables

**Table 1 nutrients-12-01676-t001:** Summary of studies on fruit and vegetable incentives as an approach for encouraging and enabling Supplemental Nutrition Assistance Program (SNAP) participants in the United States to increase the purchase and/or consumption of healthy foods.

Author	Study Objective	Incentive Benefit	Food Eligible for Incentive Redemption	Venue Type	Program Scale	Sample Size	Relevant Findings
Alaofè et al. 2017 [25]	Examine the impact of the Double Up SNAP (DUSP) farmers’ market incentive program on awareness and access to farmer’s markets, and FV purchase and consumption in Pima County, AZ.	Token/Voucher/Coupon	Unspecified FV	Farmers’ Market	One farmers’ market	353 participants	DUSP customers reported greater consumption of FV compared to non-DUSP shoppers.
Amaro and Roberts 2017 [26]	Examine characteristics (e.g., demographics, household food security) and needs of families using SNAP incentive program and evaluate incentive program usage in terms of shopping habits, food consumption patterns, and household food security.	Token/Coupon/Voucher	Unspecified FV	Farmers’ Market	One farmers’ market	143 parents	Participants reported a positive impact of incentive program use and appeared to value fresh fruits and vegetables. The majority of participants reported that the incentive enabled them to afford to shop at the farmers’ markets using their SNAP funds.
Bartlett et al. 2014 [7]	Assess the causal impact of incentive on FV consumption, and other key measures of dietary intake, by SNAP participants.	POS	Any SNAP eligible item	Farmers’ Market and Grocery Stores	130 retailers in a single metropolitan area	7500 households	Participants reported higher consumption of dark green vegetables, red/orange vegetables, and other vegetables, as well as fruits other than citrus, melons, and berries than non-Healthy Incentives Program (HIP) participants. Participants increased their consumption of vegetables more than their consumption of fruit. Increased fruit and vegetable consumption drove an increased score on the 2010 Healthy Eating Index.
Bowling et al. 2016 [27]	Examine effects of program participation on participants’ FV and soda consumption; investigate the program’s effect on WIC/SNAP budget FV expenditure patterns and use of food assistance at participating farmers’ markets, and explore the relative importance of financial (access) incentives and exposure interventions as drivers of participant enrollment and retention, as well as participants’ perceptions of barriers, support, and benefits from participation.	POS and Token/Coupon/Voucher	Fresh FV	Farmers’ Market	Six farmers’ markets in a single metropolitan area	146 households	Participants reported significantly higher vegetable consumption and lower soda consumption post implementation of the incentive program.
Dimitri et al. 2013 [28]	Examine the association between monetary incentives given for the purchase of fresh FV and fresh produce consumption among federal food assistance participants, including SNAP.	Token/Voucher/Coupon	Fresh FV	Farmers’ Market	73 farmers’ markets	1227 participants	Participants perceived increased consumption of fresh FV because they shopped at the market when offered financial incentives. Participants living in areas with low FV access, and who were also income constrained, were the most likely to perceive higher FV consumption.
Dimitri et al. 2015 [29]	Assess the effectiveness of incentives on the intake of fresh vegetables among federal food assistance program participants.	Token/Voucher/Coupon	Fresh FV	Farmers’ Market	Five farmers’ markets in three metropolitan areas	138 households	Incentives increased FV consumption overall. Groups responded to incentives differently based on level of food insecurity and education.
Durward et al. 2019 [30]	Evaluate the effect of Utah Double Up Food Bucks (DUFB) program on FV intake and food security status among SNAP participants.	Token/Voucher/Coupon	Fresh FV	Farmers’ Market	17 farmers’ markets in Utah; evaluation data collected from a sample of eight markets	138 participants	Increase in median FV consumption and percentage of participants in the incentive program who were food secure increased.
Freedman et al. 2014 [31]	Examine the influence of FV incentive on intervention program on farmers’ market revenue. The intervention included a federal monetary incentive to increase fruit and vegetable purchases at farmers’ markets for food assistance recipients.	POS	Unspecified FV	Farmers’ Market	One farmers’ market	336 participants	Incentives increased farmers’ market revenue and improved access to FV.
Lindsay et al. 2013 [32]	Examine patterns of enrollment and market visits, participants’ self-reported dietary changes while participating in the program, and the economic benefits of the program.	Token/Voucher/Coupon	Fresh FV, eggs, bread, and meat	Farmers’ Market	Five farmers’ markets in a single metropolitan area	252 participants	Incentives increased daily FV consumption and weekly FV spending.
Marcinkevage et al. 2019 [33]	Examine strengths and weaknesses of the FV prescription program implementation; gain insight into successful programming activities for FV prescriptions; assess overall effectiveness of the program in improving affordability of healthy foods among low-income patients; and assess patient satisfaction with the program.	Token/Voucher/Coupon	Fresh, Frozen, Canned, and/or Dried FV	Grocery Store	169 grocery stores in a single state	144 participants	Incentives increased FV purchase and consumption, participants reported managing their health conditions better, and an improvement in meeting nutrition, diet-related, or meal plan goals.
Olsho et al. 2015 [34]	Assess the effectiveness of incentive program in increasing access to and awareness of farmers’ markets, and increasing purchase and consumption of fruits and vegetables.	Token/Voucher/Coupon	Fresh FV	Farmers’ Market	86 farmers’ markets in a single metropolitan area	2287 participants	Health Bucks increased awareness of farmers’ markets and FV purchases. No significant change in FV consumption was detected.
Pellegrino et al. 2018 [35]	Determine FV consumption among incentive program participants and identify demographic and behavioral factors associated with higher consumption.	Token/Voucher/Coupon	Fresh FV	Farmers’ Market	Six farmers’ markets in a single metropolitan area	228 participants	Participants reported higher median FV consumption than people with similar income levels, but still below recommended levels.
Ratigan et al. 2017 [36]	Examine the factors associated with the ongoing utilization of a farmers’ market incentive program among federal food assistance participants.	Token/Voucher/Coupon	SNAP participants could use the incentive for any SNAP-eligible food, WIC participants could use the incentive for fresh produce only.	Farmers’ Market	Five farmers’ markets in a single metropolitan area	7298 participants	Increases in FV consumption and spending and improvement in perception of overall diet quality. Factors, including ethnicity, type of government assistance, age, disability status, enrolment market, season of enrolment, baseline FV serving, and perceived diet quality, affected program utilization and retention.
Rummo et al. 2019 [37]	Evaluate the effectiveness and feasibility of incentive program in grocery stores.	Token/Voucher/Coupon	Fresh FV	Grocery Store	16 grocery stores in a single metropolitan area	Unspecified number of participants	Incentives had a positive effect on FV sales but did not affect spending on sugar-sweetened beverages.
Savoie-Rosko et al. 2016 [38]	Determine whether participation in a farmers’ market incentive pilot program had an impact on food security and FV intake of participants.	Token/Voucher/Coupon	Unspecified FV	Farmers’ Market	One farmers’ market	54 participants	Incentives decreased food insecurity-related behaviors and increased intake of select FV.
Savoie Roskos et al. 2017 [39]	Identify benefits and barriers to using a farmers’ market incentive program.	Token/Voucher/Coupon	Any SNAP eligible item	Farmers’ Market	One farmers’ market	28 participants	Incentives reduced barriers associated with farmers’ market use, including cost and accessibility, and provided more spending flexibility, as well as enabled the purchase of FV that previously did not fit into budget.
Steele-Adjognon et al. 2017 [40]	Analyze how FV expenditures, expenditure shares, variety, and purchase decisions were affected by the initiation and conclusion of an FV incentive program, as well as analyze any persistent effects of the program.	Token/Voucher/Coupon	Fresh FV	Grocery Store	One grocery store	156 participants	Incentives were associated with increased vegetable spending, FV expenditure shares, and variety of FV purchased, but the effects were minimal and unsustainable without the continuation of the program. Fruit spending and FV purchase decisions were not impacted by the program.
Wetherill et al. 2017 [41]	Describe the design, implementation, and consumer response to a coupon-style intervention aimed to increase SNAP use at a farmers’ market among Temporary Assistance for Needy Families (TANF) participants.	Token/Voucher/Coupon	Unspecified FV	Farmers’ Market	One farmers’ market	254 participants	Very few participants (6.3%) redeemed the incentive coupons. Stand-alone coupon incentive programs may not be sufficient for encouraging farmers’ market use among the population using TANF. Complementary strategies, such as education, to build vegetable preparation knowledge and skills are needed.
Young et al. 2013 [42]	Determine if FV incentive program was associated with increased FV consumption and SNAP sales at farmers’ markets in low-income areas.	Token/Voucher/Coupon	Fresh FV	Farmers’ Market	22 farmers’ markets in a single metropolitan area	662 participants	Incentives were tied to increases in FV consumption and sales.

FV: fruits and vegetables, WIC: Special Supplemental Nutrition Program for Women, Infants, and Children, POS: point of sale.

**Table 2 nutrients-12-01676-t002:** Assessment of fruit and vegetable purchases in nutrition incentive programs.

Author	Assessment Method
**Survey Assessment of FV Purchases**	
Alaofè et al. (2017) [25]	Frequency of farmer’s market shopping, purchasing amount, and types of purchases were assessed by the questions: 1. “Because of Double-Up SNAP Pilot (DUSP) program rebates, is your family buying a larger amount of…?” 2. “Because of DUSP program rebates, is your family eating a greater amount of…?”, and 3. “Because of DUSP program rebates, have you or your family tried any new or unfamiliar fruits or vegetables?”
Amaro and Roberts (2017) [26]	Open-ended survey responses demonstrated that participants purchased FV at the farmers’ market because the incentive program made it affordable for them to do so. Additionally, they were asked to indicate the degree to which they agreed or disagreed with “I can afford to buy fresh fruits and vegetables”.
Bartlett et al. (2014) [7]	Specific survey items not provided but questions sought to discern general food shopping patterns and food expenditures.
Bowling et al. (2016) [27]	“How much of your family’s weekly WIC/SNAP budget is spent on FVs?”
Dimitri et al. (2013) [28]	Survey assessed questions covering five aspects: (1) frequency of shopping at farmers’ markets and the number of years receiving incentives, (2) perception of how much incentives influenced the decision to shop at the farmers’ market, (3) perception of the impact that shopping at the market with incentives had on fresh FV consumption, (4) importance of farmers’ market characteristics on the decision to shop at that market, and (5) access to the market and use of the market for fresh FV.
Lindsay et al. (2013) [32]	“How much on average do you spend on fresh fruits and vegetables per week?”
Marcinkevage et al. (2019) [33]	Perceptions of affordability, purchase of FV not previously tried.
Olsho et al. (2015) [34]	Specific survey items not provided but questions sought to discern changes in farmers’ market spending, including whether FV were purchased each visit.
Ratigan et al. (2017) [36]	Perceptions of food purchasing behavior and affordability of FV, weekly spending on FV (<$10, $10–19, $20–29, $30–39, ≥$40.)
**Interviews or Focus Groups to Assess FV purchases**	
Bartlett et al. (2014) [7]	Experiences with the program, including financial impact on the household and changes in willingness to purchase FV.
Savoie-Roskos et al. (2017) [39]	Cost and budgeting as barriers to FV purchases prior to the incentive program emerged as themes and participants noted that the program helped them overcome these barriers, citing greater spending flexibility and decreased anxiety over the cost of food.
**Sales Tracking to Assess FV Purchases**	
Bartlett et al. (2014) [7]	EBT transaction data to determine Healthy Incentive Program (HIP) incentive earnings by pilot participants, focusing on HIP-eligible purchases, the amount of incentives earned, and the percent of SNAP benefits spent on HIP-eligible purchases. Analysis of spending in different types of store, focusing on spending on targeted FV in supermarkets and superstores.
Freedman et al. (2014) [31]	Sales tracking using unique identifier for each participant; transaction data, including date of transaction, customer type (patient, staff, or community member), total cost, and payment type; comparing venue revenue trends from the previous year with those during the implementation period.
Lindsay et al. (2013) [32]	Data were collected from vendors regarding total sales each day from incentive tokens as a percentage of total sales.
Marcinkevage et al. (2019) [33]	Quarterly and yearly redemption rates, dollar amount spent on FV per incentive redeemed.
Olsho et al. (2015) [34]	Comparison of average daily SNAP sales from farmers’ markets accepting incentives with those not accepting incentives.
Ratigan et al. (2017) [36]	Records of market attendance and frequency of visits to booths where participants received incentives.
Rummo et al. (2019) [37]	FV spending as a percentage of total spending from individual transactions at grocery stores that implemented programs and that did not implement programs.
Steele-Adjognon et al. (2017) [40]	Loyalty card scanner data was acquired to assess: “FV expenditure; fruit expenditure; vegetable expenditure; FV expenditure share; FV variety; and FV purchase decision. FV expenditure is the aggregate dollar amount spent during the month on all fresh FV.”
Wetherill et al. (2017) [41]	Differences in baseline sociodemographic, predisposing, enabling, and reinforcing factors related to FV attitudes and behaviors by incentive redemption.
Young et al. (2013) [42]	Comparison of market SNAP sales from implementation period to those from previous years; incentive redemption rates.

**Table 3 nutrients-12-01676-t003:** Assessment of fruit and vegetable consumption in nutrition incentive programs.

Author	Description of Assessment Method
Alaofè et al. (2017) [25]	FV consumption frequency measured using Behavioral Risk Factor Surveillance System FV module.
Bartlett et al. (2014) [7]	24-h dietary recall interviews at multiple points in implementation period and followed up by focus groups, which included discussion of impact on FV consumption. Surveys on FV consumption (frequency and quantity) using Eating at America’s Table Study (EATS) Fruit and Vegetable Screener.
Bowling et al. (2016) [27]	Survey questions including “On an average day, how many times do you have a vegetable to eat?” and “On an average day, how many times do you have a fruit to eat?”
Dimitri et al. (2015) [29]	National Health and Nutrition Examination Survey food frequencyquestionnaire: Number of times vegetables were consumed in the last six months, daily and weekly serving of FV.
Dimitri et al. (2013) [28]	Specific survey items not provided, but assessed participant perception that fresh FV consumption increased or did not increase.
Durward et al. (2019) [30]	FV consumption frequency measured using Behavioral Risk Factor Surveillance System FV module.
Lindsay et al. (2013) [32]	“On average, how many servings of fruits and/or vegetables do you usually eat each day?” and “In general, how healthy would you say your overall diet is?”
Marcinkevage et al. (2019) [33]	Survey included questions related to participant perceived improvement in the consumption of healthy foods, including FV, and perceived health benefit prescriptions (trying new FV, eating more FV, increases in FV consumption by family members.)
Olsho et al. (2015) [34]	New York City Community Health Survey: “total servings of fruits and vegetables eaten on the previous day” and “consumption today vs. consumption one year ago”; interviews included questions about the consumption of FV from farmers’ markets.
Pellegrino et al. (2018) [35]	FV consumption frequency measured using Behavioral Risk Factor Surveillance System FV module.
Ratigan et al. (2017) [36]	Survey regarding number of servings of FV consumed daily, rank overall dietary quality (very healthy, healthy, average, unhealthy, very unhealthy.)
Savoie-Roskos et al. (2016) [38]	FV consumption frequency measured using Behavioral Risk Factor Surveillance System FV module.
Savoie-Roskos et al. (2017) [39]	Interview: “How does your diet now compare to your diet before the study?”
Young et al. (2013) [42]	“Since becoming a customer at this market, do you eat more, less, or the same amount of fruits and vegetables?”

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
