# Peer review of "Fruit and Vegetable Incentive Programs for Supplemental Nutrition Assistance Program (SNAP) Participants: A Scoping Review of Program Structure"

_nutrients, 2020, doi:10.3390/nu12061676_

Round 1

Reviewer 1 Report

This paper was very well written.  Only a few comments that should be easily addressed, as follows:

Abstract

Line 18 and Line 23: It is not grammatically appropriate to begin a sentence with a numeral “11/19” and “18/19”

Introduction

Line 54:  regarding the statement, “However, key constituencies ranging from members of U.S. Congress to hunger relief organizations have rejected these proposals for reasons including concerns about limiting participants’ ability to exercise autonomy in food choice and administrative burdens.”  Another important issue is food deserts and restrictions possibly worsening food insecurity if SNAP cannot be used to purchase foods that are locally available. I think this needs to be named as another important factor as to why SNAP purchasing restrictions is not welcomed by anti-hunger advocates.

Line 92: Was a librarian consulted in the selection of search terms?  My understanding is PRISMA guidelines (and standard systematic review practice) encourage this.  Also, two reviewers typically independently review the abstracts from the search results before deciding on the final selection.  Was this step done, and what was the agreement between reviewers?  How were discrepancies resolved? If this step was not done, I think the authors should complete this step (as it would only take an hour or so) to confirm their final selection of articles was appropriate.

Discussion

Line 346: the Wetherill et al study was not representative of the standard SNAP population as it only included TANF participants.  These are very low-income individuals with incomes that will be less than an average SNAP participant.  It would be good to clarify that this study was limited to a sub-population of SNAP users, TANF participants, who may require additional support to address FV disparities compared to general SNAP users.

It would be nice to see a sentence or two about many of the nutrition outcome measures not being appropriate for assessing changes in dietary intake as the result of an intervention.  To properly assess for changes, a single 24-hr food recalls should be used as the gold standard. Future studies should strive to employ validated dietary measures.

Reviewer 2 Report

Dear authors,

Thank you for the opportunity to review this manuscript. 

Please see comments below:

General/abstract

  • Avoid the use of an acronym and numbers at the start of sentences
  • The authors affiliations appear to be incomplete
  • Suggest reviewing key words to expand these
  • Line 12 - include reference to scoping review
  • There is a limited overview of methods in the abstract

Introduction

  • Overall the introduction could be improved, particularly to highlight the rationale for the study and to provide a complete background for those readers not familiar with SNAP.
  • The first paragraph is very short/incomplete
  • The first two paragraphs appear slightly disjointed, i.e there is reference to 'critical' in line 37, but this is not clear in the context of the sentence. Line 36 would benefit from clarification on what the differential is prior to this. 
  • It would be of benefit to provide an overview of SNAP for non-American readers
  • Line 69 needs a reference
  • Line 74 would benefit from more references

Materials and methods

  • This section requires improvement.
  • Reference 17 (Tricco et al) reports on the PRISMA Extension for reporting scoping reviews, and is not the most appropriate reference for the method undertaken. It would be worth considering Arskey & O'Malley for methods.
  • The method is very brief, at the very least it should include reference to the stages completed, including development of the research question, how the search terms were developed (including any assistance from Librarians), that inclusion/exclusion criteria were pre-defined, how the process of removing duplicates was handled i.e. were the manuscripts downloaded to a reference manager or similiar, and what was the standardised data abstraction tool?
  • Would be good to see justification for scoping review process (as opposed to other reviews)

Results

  • The reference to Table 1 could be clearer (perhaps remove sentence form and add in brackets at end of previous sentence).
  • Some of the data presented maybe clearer to readers in table form.
  • Suggest reconsidering the title of this section. There appears to be discussion throughout, but a usual results section does not present this. 
  • Some writing could be tightened up to help readability and repetition.

Conclusions/recommendations

  • Quite clear and align to the study
  • Line 391 - demonstrations ?

Thank you,

Round 2

Reviewer 2 Report

Dear authors,

Thank you very much for the opportunity to review the revised version of this manuscript. 

I still have concerns about the methods section of this manuscript. These  are as follows:

  • There is no statement in the manuscript providing rationale for the ScR - see PRISMA ScR Checklist introduction/rationale.
  • Tricco et al. has been used by other authors to justify the ScR process, however the Tricco et al. paper is based on reporting of ScR. If the authors make reference to this, a completed PRISMA ScR reporting checklist should be provided. There should also be reference to the methodology used. 
  • There is no reference to a protocol document, which is used in both  key ScR guiding documents by JBI and Arskey and O'Malley. Even if JBI/Arskey and O'Malley were not used, a protocol is included on the PRISMA ScR checklist, so should be used if this is guiding the ScR process. 
  • The research question has still not been identified (who/how this was developed)
  • There is no differentiation between title/abstract and full screening in the manuscript.
  • There is limited information on the data extraction process, particularly in reference to what is required on the PRISMA ScR checklist. 

The rest of the revisions appear to meet the initial queries. 

Thank you,
